# Inflammatory Biomarkers in the Diagnosis and Prognosis of Rheumatoid Arthritis–Associated Interstitial Lung Disease

**DOI:** 10.3390/ijms24076800

**Published:** 2023-04-05

**Authors:** Natalia Mena-Vázquez, Francisco Javier Godoy-Navarrete, Jose Manuel Lisbona-Montañez, Rocío Redondo-Rodriguez, Sara Manrique-Arija, José Rioja, Arkaitz Mucientes, Patricia Ruiz-Limón, Aimara Garcia-Studer, Fernando Ortiz-Márquez, Begoña Oliver-Martos, Laura Cano-García, Antonio Fernández-Nebro

**Affiliations:** 1Instituto de Investigación Biomédica de Málaga (IBIMA)-Plataforma Bionand, 29010 Málaga, Spain; 2UGC de Reumatología, Hospital Regional Universitario de Málaga, 29009 Málaga, Spain; 3UGC de Reumatología, Hospital Universitario de Jaén, 23007 Jaén, Spain; 4Departamento de Medicina y Dermatología, Universidad de Málaga, 29010 Málaga, Spain; 5UGC de Endocrinología y Nutrición, Hospital Clínico Virgen de la Victoria, 29010 Málaga, Spain; 6CIBER Fisiopatología de la Obesidad y Nutrición (CIBEROBN), Instituto de Salud Carlos III, 28029 Madrid, Spain; 7UGC de Neurociencias, Hospital Regional Universitario de Málaga, 29010 Málaga, Spain

**Keywords:** rheumatoid arthritis, interstitial lung disease, biomarkers, cytokines, morbidity, inflammation

## Abstract

This study aimed to identify inflammatory factors and soluble cytokines that act as biomarkers in the diagnosis and prognosis of rheumatoid arthritis-associated interstitial lung disease (RA-ILD). We performed a nested prospective observational case–control study of patients with RA-ILD matched by sex, age, and time since the diagnosis of RA. All participants underwent pulmonary function testing and high-resolution computed tomography. ILD was defined according to the criteria of the American Thoracic Society/European Respiratory Society; the progression of lung disease was defined as the worsening of FVC > 10% or DLCO > 15%. Inflammation-related variables included the inflammatory activity measured using the DAS28-ESR and a multiplex cytokine assay. Two Cox regression models were run to identify factors associated with ILD and the progression of ILD. The study population comprised 70 patients: 35 patients with RA-ILD (cases) and 35 RA patients without ILD (controls). A greater percentage of cases had higher DAS28-ESR (*p* = 0.032) and HAQ values (*p* = 0.003). The variables associated with RA-ILD in the Cox regression analysis were disease activity (DAS28) (HR [95% CI], 2.47 [1.17–5.22]; *p* = 0.017) and high levels of ACPA (HR [95% CI], 2.90 [1.24–6.78]; *p* = 0.014), IL-18 in pg/mL (HR [95% CI], 1.06 [1.00–1.12]; *p* = 0.044), MCP-1/CCL2 in pg/mL (HR [95% CI], 1.03 [1.00–1.06]; *p* = 0.049), and SDF-1 in pg/mL (HR [95% CI], 1.00 [1.00–1.00]; *p* = 0.010). The only variable associated with the progression of ILD was IL-18 in pg/mL (HR [95% CI], 1.25 [1.07–1.46]; *p* = 0.004). Our data support that the inflammatory activity was higher in patients with RA-ILD than RA patients without ILD. Some cytokines were associated with both diagnosis and poorer prognosis in patients with RA-ILD.

## 1. Introduction

Rheumatoid arthritis (AR) is a chronic inflammatory disease that mainly affects the joints. Without treatment, patients with RA experience progressive joint deterioration, which, in turn, can lead to disability and reduced quality of life. RA often affects organs and systems outside the musculoskeletal system, including the lungs [1]. RA-associated interstitial lung disease (RA-ILD) is the most common lung manifestation and can result in high morbidity and mortality [2,3]. Early treatment can help to reduce the risk of the progression of RA-ILD [4]. This requires biomarkers that can identify patients who are at risk of RA-ILD and disease progression.

No useful serum biomarkers are currently available for the diagnosis of RA-ILD, although various candidates have been evaluated. The protein Krebs von den Lungen 6 (KL-6) in serum has been investigated for the diagnosis of ILD and has been associated with systemic inflammatory diseases [5]. The highest KL-6 values have a specificity of around 90%, although sensitivity is lower [6]. Promising results have also been reported for serum levels of extracellular matrix metalloproteinase 7 [7,8,9], interferon-gamma–induced protein 10 (IP-10, also known as C-X-C motif chemokine ligand 10 [CXCL10]) [7], and the 90- and 70-kDa heat shock proteins [10]. None of these proteins are yet available in clinical practice. Similarly, they have not been proven to have a greater predictive value than the anti–citrullinated peptide antibody (ACPA) [11,12]. Other studies have described that elevated uric acid levels, D-dimer, and tumor markers may be diagnostic markers in RA-ILD [13,14].

RA was characterized by the dysregulation of innate and adaptive immune systems, in which cytokines play a major role. Given that cytokines contribute to chronic inflammation and joint destruction, there are many reasons why they should be assessed in patients with RA-ILD. First, they can provide valuable information on all phases of the pathogenesis of RA through their role in promoting autoimmunity, maintaining chronic synovitis, and leading to the destruction of neighboring joint tissue [15,16,17,18]. Second, cytokine levels have been shown to be elevated in many patients with ILD, thus suggesting that they may also be involved in the development and progression of this condition. In fact, ILD is characterized by alveolar and interstitial inflammation, and the role of cytokines in the pathogenesis of ILD has been known for some time [19,20]. Nevertheless, cytokines have received little attention in RA-ILD, with the result that our knowledge is very limited. The production of TNF and IL-6 by alveolar macrophages has been shown to be increased in patients with RA-ILD, although this determination was performed in only eight patients and in bronchoalveolar lavage fluid [21]. A more recent study revealed higher titers of the IL-1 alpha antibody in the serum of patients with RA-ILD than in those without ILD [22,23]. Similarly, TGF beta1 [24] and IL-18 [8] have been associated with the diagnosis and severity of RA-ILD [24], and the level of circulating MMP-3 increased the diagnostic accuracy for ILD in RA patients [25]. Lastly, understanding the role of cytokines in RA-ILD could help us to identify potential therapeutic targets that might improve outcomes in affected patients. Interleukins are involved in pro and anti-inflammatory responses due to their interaction with a wide range of receptors, e.g., Toll-like receptors (TLRs). Interleukins and TLRs are involved in cancers along with infectious and autoimmune diseases [26].

While interest in ILD has grown in the last 10 years, and studies are being performed to gain a greater insight into the clinical characteristics and risk factors associated with RA-ILD, our knowledge remains very limited [2,11]. Given the absence of a clinically validated biomarker to improve the diagnosis of early RA-ILD and/or to predict the response to treatment, this disease is usually diagnosed late and is based on multiple complex diagnostic tests. Therefore, the objectives of our study are as follows: (1) to identify soluble cytokine biomarkers in patients with RA-ILD; and (2) to describe their association with other factors involved in the progression of lung disease.

## 2. Results

### 2.1. Baseline Characteristics

The study population comprised 70 patients: 35 patients with RA-ILD (cases) and 35 RA patients without ILD (controls) (Figure 1). Somewhat more than half were men (57.1%), and the mean (SD) age was 68.1 (8.3) years. The median (IQR) follow-up of RA was 139.8 (79.5–220.6) months. All the controls had a normal HRCT scan, and none had a history of pulmonary symptoms. Table 1 shows the baseline characteristics of both groups. As we can see, both groups were well balanced in most of the characteristics of RA, except that more patients in the ILD group had positive autoantibody titers.

All the participants received a DMARD (Table 1). While there were no differences in terms of the percentage of csDMARDs and bDMARDs, the cases less frequently took methotrexate (*p* = 0.040) and anti-TNF agents (*p* = 0.041) and more frequently took hydroxychloroquine (*p* = 0.010) and abatacept (*p* = 0.004). Only 4 patients with RA-ILD received mycophenolate mofetil (*p* = 0.032), and 1 patient took antifibrotic medication combined with mycophenolate mofetil and abatacept.

According to the protocol, the HRCT scan revealed lung involvement in all the cases and none of the controls. The most common histopathologic patterns visible on the HRCT were UIP in 29 patients (82.9%) and NSIP in 6 (17.1%). As was to be expected, the cases had lower mean PFT values than the controls (Table 1). The PFT values in these cases revealed a reduced DLCO in 29 patients (85.3%) and reduced FVC in 28 (80.0%).

### 2.2. Inflammation

Table 2 shows the disease activity parameters for the cases and the controls. The cases had higher DAS28-ESR values (*p* = 0.032), a higher number of swollen joints (*p* = 0.040), and poorer physical function according to the HAQ (*p* = 0.003). As for cytokines, the cases generally had higher values of IL-1 alpha, IL-6, IL-18, MCP-1/CCL2, MIP1beta (CCL4), and SDF-1 alpha than the controls (Table 2). In patients with moderate-high inflammatory activity, RA ILD patients had higher values of IL-18 (Appendix A).

Table 3 and Figure 1 show the results of the univariate Cox regression analysis (adjusted for the time since the diagnosis of RA), which was performed to identify the factors associated with RA-ILD. The variables that were independently associated with ILD were moderate-high disease activity according to the DAS28-ESR (β = 0.914; *p* = 0.017) and high levels of ACPA (β = 1.064; *p* = 0.014), IL-18 in pg/mL (β = 0.091; *p* = 0.044), MCP-1/CCL2 in pg/mL (β = 0.031; *p* = 0.049), and SDF-1 alpha in pg/mL (β = 0.081; *p* = 0.010).

### 2.3. Progression of Lung Disease

Thirteen patients (37.1%) fulfilled the criteria for the progression of lung disease, 20 (57.1%) of these were for stabilization, and 2 (5.7%) were for improvement, with a mean (SD) follow-up of ILD for 66.1 (47.2) months. At the cut-off, the mean PFT values had fallen significantly compared with the date of the diagnosis of RA-ILD for FVC (mean [SD], 63.0 [17.1] vs. 69.4 [14.8] mg/L; *p* = 0.001), DLCO (mean [SD], 62.9 [15.0] vs. 68.9 [14.4] mg/L; *p* < 0.001), and FEV1 (mean [SD], 68.7 [15.9] vs. 75.1 [13.8] mg/L; *p* = 0.003) (Appendix A).

As can be seen in Table 4, lung disease progressed in 13 of the 35 patients with RA-ILD (37.1%) during follow-up. Those whose lung disease progressed more frequently had high ACPA titers (*p* = 0.041). While progression was more common in men and with the UIP pattern, these differences were not significant (Appendix A). As for the cytokine profile, patients with RA-ILD and the progression of lung disease had higher values of IL-1 alpha, IL-18, and MCP-1/CCL2 than those whose disease did not progress (Table 4). Appendix A shows the results of an alternative Cox regression analysis that was adjusted for the duration of ILD, with the dependent variable progression of RA-ILD. The only variable that was associated with the progression of RA-ILD in this model was IL-18 in pg/mL (β = 0.227; *p* = 0.004) (Figure 2).

## 3. Discussion

Several biomarkers that could prove potentially useful in the diagnosis and follow-up of RA-ILD have been identified. However, the role of cytokines in RA-ILD has received scant attention. The present study compared cytokine levels and inflammatory activity in patients with RA-ILD and patients with RA but not ILD. In addition to the determination of cytokines, all patients underwent complete PFT. Given that patients with RA-ILD underwent complete PFT since the diagnosis of ILD, we were also able to evaluate the association between cytokines and the progression of lung disease.

In line with this approach, our findings show that inflammatory activity according to the DAS28-ESR is more pronounced in patients with RA-ILD, and these patients more frequently had high ACPA titers. They also showed differences in the overexpression of cytokines, specifically MCP-1/CCL2, SDF-1 alpha, and IL-18. In this sense, other studies have also found that patients with RA-ILD have more pronounced joint disease activity than RA patients without ILD [27,28]. In fact, one study showed that patients with moderate-to-highly active RA (DAS28) were at twice the risk of ILD compared to patients with RA in remission or with low disease activity after adjusting for sex, smoking, disease duration, and serology status [29]. Some studies have tried to explain how active joint involvement in RA can lead to continuously high levels of systemic inflammation that can, in turn, cause inflammation and the development of RA-ILD [30]; therefore, joint and lung inflammation may be related [31,32]. Moreover, the presence of ACPA, especially at higher titers, is another factor that is widely known to be associated with RA-ILD [33]. In fact, to date, none of the biomarkers under study have been proven to have a higher predictive value for ILD than ACPA [2,11].

As for the cytokines we found to be associated with ILD, MCP-1/CCL2 was involved both in RA and in ILD. On the one hand, in patients with RA, the signaling pathway for chemokines (CCL4/CCR5/c-Jun and c-Fos/CCL2) was involved in the expression of CCL2, which could lead to the chronic inflammation associated with RA [34]. On the other hand, some studies have shown an association between interstitial lung involvement and levels of MCP-1/CCL2 in patients with systemic sclerosis [35]. A recent study suggested that MCP-1/CCL2 was a predominant source for the replenishment of lung macrophages during lung remodeling [36]. Elsewhere, lung macrophages induced by MCP-1/CCL2 in patients with COVID-19 and lung involvement have been shown to share a transcriptional phenotype with macrophages stimulated by TNF alpha and IFN gamma [37]. Furthermore, SDF-1 alpha is a class of chemokine with multiple cellular effects and a potent angiogenic effect [38]. Patients with ILD associated with autoimmune diseases have been reported to have significantly higher SDF-1 alpha values than patients with idiopathic pulmonary fibrosis, although, in both cases, they have increased [39]. In addition, some cytokines may play a role in RA. SDF-1 alpha could play a role in the pathogenesis of RA since it activates fibroblast-like synoviocytes, stimulates angiogenesis, and degrades the cartilage matrix with the release of metalloproteinases [40]. However, no correlation has been observed with disease activity or severity [38]. Nevertheless, Margaritopoulos et al. [41] found that the development of pulmonary fibrosis in RA can be considered the main signal for stem cell migration to the damaged lung via the SDF-1 alpha/CXCR4 axis.

In the present study, we also found that IL-18 values were higher in cases than in the controls. Similarly, this was the only cytokine associated with the progression of RA-ILD. IL-18 is a member of the IL-1 cytokine superfamily and is produced predominantly by macrophages. Data from animal studies show that IL-18 can lead to pulmonary inflammation [42]; in humans, IL-18 levels increased in patients with idiopathic pulmonary fibrosis [43], patients with ILD-associated inflammatory myopathy [44], and patients with RA-ILD [8]. Similarly, IL-18 levels were associated with the pathogenesis of RA and had high biologic activity in synovial fluid and the sera of patients with RA [26,45]. In this sense, Matsuo et al. recently showed that patients with RA-ILD had higher IL-18 values than RA patients without ILD; IL-8 was also associated with ILD, independently of inflammatory factors [8]. This IL-18–mediated effect could be due to various mechanisms, i.e., a key role in the polarization of Th1 cells, the production of inflammatory cytokines by different cell strains in innate and acquired immunity, and the differentiation of Th17 cells [46]. While further studies are necessary to determine the exact role of this cytokine in RA-ILD, we might think that its association with greater inflammation and severity in RA is associated with a poorer outcome of ILD in affected patients.

Our study is subject to a series of limitations. First, the analysis of inflammation and cytokines was performed ad hoc, thus preventing us from confirming a causal relationship with other variables. However, a key strength of our study was that both groups of patients were studied using HRCT and PFT, thus enabling us to clearly identify cases and the controls and assess the differences between them. Furthermore, the prospective follow-up of the RA-ILD cohort enabled us to study the progression of lung disease and its association with cytokines and inflammation. It is also noteworthy for all the participants who received DMARDs that these drugs directly affected cytokines and re-established the inflammatory process. Nevertheless, since both the cases and the controls were treated with DMARDs, this effect applied to all the participants included. Furthermore, we did not include a healthy control group without RA since the main objective was to identify soluble cytokine biomarkers in patients with RA-ILD and the progression of lung disease; performing HCRT in healthy controls could be more controversial. Lastly, our sample may have been too small to detect more robust differences and more clinical or laboratory factors. However, cases differed from the controls in inflammatory parameters, and some of the cytokines studied.

## 4. Materials and Methods

### 4.1. Study Design and Participants

We performed an observational case–control study nested in a single-center prospective cohort of patients with RA from Hospital Regional Universitario de Málaga (HRUM), Malaga, Spain. The study was approved by the Research Ethics Committee of HRUM (2627-N-21). All patients gave their written informed consent before participating.*Cases*

The case group comprised patients with established RA from the initial cohort of HRUM [47] who were clinically diagnosed with ILD. These patients formed part of an ongoing prospective sub-cohort that was started in 2015. Patients from the sub-cohort who attended the outpatient clinic for check-ups between January 2020 and 2021 were invited to participate as cases. The inclusion criteria were age ≥16 years, a diagnosis of RA based on the 2010 classification criteria of the American College of Rheumatology/European League Against Rheumatism (ACR/EULAR 2010) [48], ILD diagnosed according to the criteria of the American Thoracic Society/European Respiratory Society [49], and treatment with disease-modifying antirheumatic drugs (DMARDs) for at least one year. We excluded patients with concomitant inflammatory or rheumatic disease other than RA (except secondary Sjögren syndrome) and pregnant women.*Controls*

The control group comprised patients with clinically significant RA without ILD who were consecutively selected from all other patients with RA in the prospective RA cohort of HRUM and followed the same procedure as the cases in [47]. Each case was matched with control for sex, age (±3 years), and time since diagnosis (±24 months). The inclusion criteria were the same as those of the cases, except that the controls could not have pulmonary symptoms (cough or dyspnea). In addition, they had to have normal pulmonary function test (PFT) results and a normal high-resolution computed tomography (HRCT) scan. Figure 3 shows a flow chart of the process used to select the cases and controls.

### 4.2. Protocol

All participants were managed according to a pre-established protocol for data collection after signing an informed consent document. A blood sample was taken to measure the levels of cytokines and other inflammatory parameters from all patients at inclusion. All patients with RA-ILD underwent systematic PFT and HRCT at a diagnosis of ILD, at years 2 and 5 if they remained stable, and at any other visit if clinically necessary. At inclusion, all participants underwent PFT and HRCT, with this date being considered the cut-off. The approach for reading the HRCT scan and the methodology have been published elsewhere [1,50,51]. Controls with pulmonary symptoms or abnormal HRCT findings were excluded from the study.

### 4.3. Main Outcomes

The 2 main outcomes were [1] the presence of RA-ILD and [2] the progression of lung disease in patients with RA-ILD. RA-ILD was confirmed by the presence of respiratory symptoms, including any identifiable ILD on the HRCT scan and/or lung biopsy. ILD was classified according to the standard criteria of the American Thoracic Society/European Respiratory Society International Multidisciplinary Consensus Classification of the Idiopathic Interstitial Pneumonias, which recognizes 3 patterns: nonspecific interstitial pneumonia (NSIP), usual interstitial pneumonia (UIP), and other patterns (bronchiolitis obliterans, organizing pneumonia, lymphoid interstitial pneumonia, and mixed patterns) [52].

As for the progression of lung disease, we defined 3 stages: [1] progression (worsening of forced vital capacity [FVC] > 10% or the diffusing capacity of the lungs for carbon monoxide [DLCO] > 15%); [2] non-progression (stability or improvement in FVC ≤ 10% or DLCO ≤ 15%); and [3] improvement (increase in FVC > 10% or DLCO > 15%) [11]. PFT included complete spirometry, which was expressed as a percent that was predicted and adjusted for age, sex, and height. Abnormal FVC was defined as ≤80% of those predicted. DLCO was evaluated using the single-breath method and was considered abnormal if its value was ≤80%.

The secondary outcome was the radiologic progression on HRCT, defined as an increase of 20% or more in the presence and extension of ground-glass opacities, reticulation, honeycombing, low attenuation, centrilobular nodules, other nodules, emphysema, or consolidation compared with the baseline HRCT scan.

### 4.4. Other Variables

Inflammation-related variables included data on inflammatory activity based on the multiplex cytokine assay and clinical activity indices. Clinical inflammatory activity was evaluated for all participants using the 28-joint Disease Activity Score with an erythrocyte sedimentation rate (DAS28-ESR) (continuous, range 0–9.4). The results of the DAS28-ESR were stratified as follows: [1] high activity, >5.1; [2] moderate activity, 3.2–5.1; [3] low activity, 2.6–3.2; and [4] remission, <2.6 [53]. The multiplex cytokine assay was run using the Bio-Plex Suspension Array System 200 (Bio-Rad Laboratories, Hercules, CA, USA) to quantify plasma cytokine and chemokine concentrations with an immunoassay panel (ProcartaPlex) and an appropriate standard plasma diluent kit (Invitrogen, Thermo Fisher Scientific, Waltham, MA, USA) [54,55]. The analysis included cytokine levels that intervened in the Th1/Th2 function (GM-CSF, IFN gamma, IL-1 beta, IL-2, IL-4, IL-5, IL-6, IL-8, IL-12p70, IL-13, IL-18, TNF alpha) and Th9/Th17/Th22/Treg function (IL-9, IL-10, IL-17A [CTLA-8], IL-21, IL-22, IL-23, IL-27), inflammatory cytokines (IFN alpha, IL-1 alpha, IL-7, IL-15, IL-31, TNF beta), and chemokines (eotaxin [CCL11], GRO alpha [CXCL1], IP-10 [CXCL10], monocyte chemoattractant protein 1 [MCP-1]/CCL2, MIP-1 alpha [CCL3], MIP-1 beta [CCL4], RANTES [CCL5], and stromal cell–derived factor 1 alpha [SDF-1 alpha]).

Severity variables included the presence of autoantibodies and their titers. The rheumatoid factor (RF) was considered positive if >10 IU/mL, ACPA was considered positive if >20 IU/mL, and high ACPA titer if >340 IU/mL. The presence of radiologic erosions and the Health Assessment Questionnaire (HAQ) score were also taken into account [56]. The duration of symptoms of RA was recorded, as was the delay between the onset of symptoms and diagnosis. Moreover, epidemiologic variables were collected at the cut-off, as follows: sex, race, body mass index (BMI [weight/height in m^2^]), history of smoking (active smokers, ex-smokers, and nonsmokers), and socioeconomic and educational level. As for treatment, we recorded conventional synthetic DMARDs (csDMARDs), biologic DMARDs (bDMARDs), other immunosuppressants, corticosteroids at the cut-off, and previous treatment.

### 4.5. Statistical Analysis

We performed a descriptive analysis of the main variables. Qualitative variables were expressed as the absolute number and percentage, and quantitative variables as the mean and standard deviation (SD) or as the median and interquartile range (IQR) depending on the normality of the distribution (Kolmogorov–Smirnov test). The χ^2^ test and *t* test or Mann–Whitney test were used to compare the main characteristics between cases and controls, as well as between cases with RA-ILD with and without the progression of lung disease at the cut-off. A paired *t*-test or Wilcoxon test, as applicable, was performed to compare PFT findings at the onset of RA-ILD and at the cut-off in patients with RA-ILD. Finally, we ran 2 stepwise Cox regression analyses (Wald), one to identify the severity and inflammatory factors associated with RA-ILD adjusted for the time since diagnosis of RA and another to determine the factors associated with the progression of RA-ILD when adjusted for the time since the diagnosis of RA-ILD. The factors that we included in this model were those that were significant in the univariate analysis. The multicollinearity of the independent variables was checked using the Pearson correlation coefficient. The analysis was performed with IBM-SPSS Statistics, Version 28.

## 5. Conclusions

In conclusion, values for inflammatory activity and ACPA were higher in patients with RA-ILD than RA patients without ILD. Some cytokines, for example, MCP-1/CCL2 and SDF-1 alpha were associated with the diagnosis of RA-ILD, and IL-18 levels were associated with a diagnosis of RA-ILD and a more marked progression of lung disease. If validated, these cytokines could be potential diagnostic and prognostic markers for RA-ILD disease and, therefore, could contribute to the early identification of patients with high morbidity and mortality. Further studies are necessary to validate these results.

## Figures and Tables

**Figure 1 ijms-24-06800-f001:**
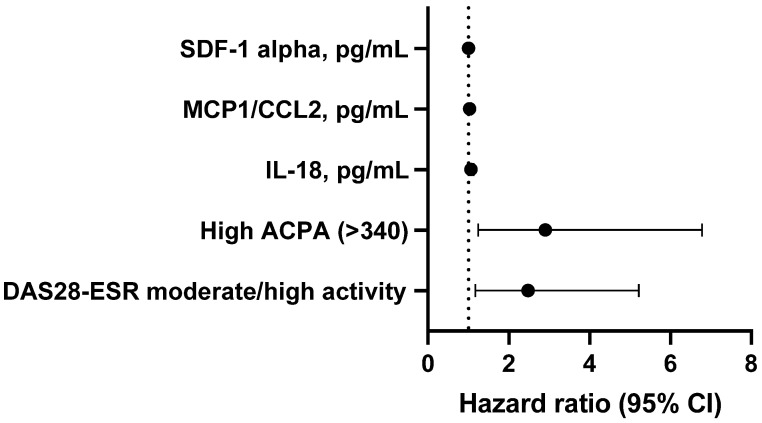
Cox regression analysis.

**Figure 2 ijms-24-06800-f002:**
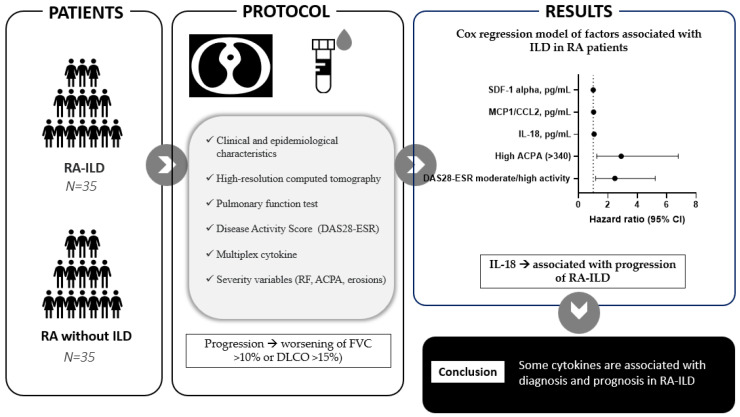
Methods and results.

**Figure 3 ijms-24-06800-f003:**
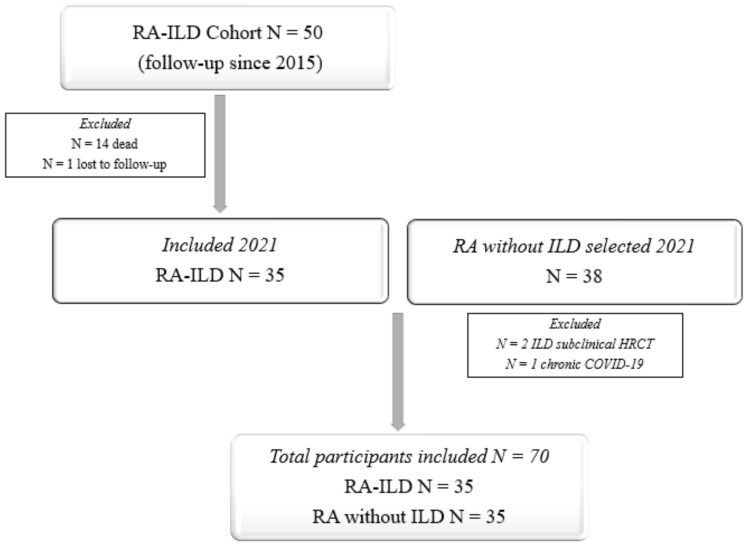
Flow diagram of patient selection into the study.

**Table 1 ijms-24-06800-t001:** Baseline characteristics of the study population.

VARIABLE	RA-ILD N = 35	RA without ILD N = 35	*p* Value
Epidemiological characteristics			
Age, years, mean (SD)	69.7 (9.3)	66.6 (7.0)	0.130
Male sex; n (%)	20 (57.1)	20 (57.1)	1.000
Clinical-analytical characteristics			
Smoking history			0.760
Nonsmokers, n (%)	17 (48.6)	18 (51.4)	
Ex-smokers, n (%)	10 (28.6)	8 (22.9)	
Active smokers, n (%)	8 (22.9)	9 (25.7)	
Time with RA, months, median (IQR)	149.8 (93.3–245.5)	133.7 (67.8–204.2)	0.384
Time with ILD, months, mean (SD)	66.1 (47.2)	-	-
RF+ (>10), n (%)	33 (94.3)	31 (88.6)	0.393
High RF (>60), n (%)	24 (68.6)	17 (48.6)	0.089
ACPA+ (>20), n (%)	32 (91.4)	31 (88.6)	0.690
ACPA titer, median (IQR)	562.0 (150.0–1305.0)	283.0 (41.0–816.0)	0.073
High ACPA titer (>340), n (%)	22 (63.0)	14 (40.0)	0.039
Radiographic erosions, n (%)	21 (60.0)	19 (55.6)	0.705
Current treatment			
csDMARD, n (%)	28 (80.0)	33 (94.3)	0.074
Methotrexate, n (%)	19 (54.3)	27 (77.1)	0.040
Leflunomide, n (%)	3 (8.6)	6 (17.1)	0.284
Sulfasalazine, n (%)	2 (5.7)	2 (5.7)	1.000
Hydroxychloroquine, n (%)	6 (17.1)	0 (0.0)	0.010
Mycophenolate, n (%)	4 (11.4)	0 (0.0)	0.032
bDMARD, n (%)	22 (62.9)	18 (51.4)	0.334
Anti-TNF, n (%)	4 (11.4)	11 (31.4)	0.041
Tocilizumab, n (%)	3 (8.6)	2 (5.7)	0.643
Abatacept, n (%)	13 (37.1)	3 (8.6)	0.004
Rituximab, n (%)	2 (5.7)	0 (0.0)	0.151
JAK inhibitor, n (%)	0 (0.0)	2 (5.7)	0.151
Corticosteroids, n (%)	22 (62.9)	6 (17.1)	0.001
Nintedanib, n (%)	1 (2.9)	0 (0.0)	0.314
Dose of corticosteroid, mg/d, mean (SD)	7.9 (4.0)	5.0 (0.0)	0.135
Pulmonary function tests			
FVC < 80%, n (%)	28 (80.0)	5 (14.3)	<0.001
FVC predicted (%), mean (SD)	63.0 (17.1)	83.4 (4.4)	<0.001
FEV1 < 80%, n (%)	23 (67.6)	5 (14.3)	0.001
FEV1 predicted (%), mean (SD)	68.7 (15.9)	84.0 (11.5)	<0.001
DLCO < 80%, n (%)	29 (85.3)	3 (8.6)	<0.001
DLCO-SB predicted (%), mean (SD)	61.0 (15.2)	85.9 (7.9)	<0.001
HRCT pattern			
UIP, n (%)	29 (82.9)	0 (0.0)	<0.001
NSIP, n (%)	6 (17.1)	0 (0.0)	<0.001

Abbreviations: RA: rheumatoid arthritis; ILD: interstitial lung disease; SD: standard deviation; RF: rheumatoid factor; ACPA: anti–citrullinated peptide antibody; csDMARD: conventional synthetic disease-modifying antirheumatic drug; bDMARD: biologic disease-modifying antirheumatic drug; FVC: forced vital capacity; FEV_1_: forced expiratory volume in the first second; DLCO: diffusing capacity of the lung for carbon monoxide; HRCT: high-resolution computed tomography; UIP: usual interstitial pneumonia; NSIP: nonspecific interstitial pneumonia.

**Table 2 ijms-24-06800-t002:** Cytokine and inflammatory activity profile of 35 patients with RA-ILD and 35 RA patients without ILD.

Variable	RA-ILD N = 35	RA without ILD N = 35	*p* Value
Physical function and activity indices			
DAS28-ESR, mean (SD)	3.1 (0.9)	2.6 (0.9)	0.032
Remission-low disease activity, n (%)	19 (54.3)	27 (77.1)	0.044
Moderate-high activity, n (%)	16 (45.7)	8 (22.9)	0.044
Number of painful joints, median (IQR)	0.0 (0.0–1.0)	0.0 (0.0–1.0)	0.792
Number of swollen joints, median (IQR)	0.0 (0.0–1.0)	0.0 (0.0–0.0)	0.040
HAQ, mean (SD)	1.2 (0.6)	0.8 (0.6)	0.003
Cytokines			
Eotaxin (CCL11), pg/mL, median (IQR)	14.9 (11.2–22.9)	13.1 (8.2–18.1)	0.185
GM-CSF, pg/mL, median (IQR)	5.5 (3.4–10.7)	5.3 (3.6–8.8)	0.853
GRO alpha (CXCL1), pg/mL, median (IQR)	0.3 (0.2–0.4)	0.3 (0.2–0.4)	0.177
IFN gamma, pg/mL, median (IQR)	0.9 (0.5–1.6)	0.8 (0.6–1.3)	0.294
IL-1 alpha, pg/mL, median (IQR)	1.0 (0.2–5.3)	0.2 (0.1–1.4)	0.043
IL-1 beta, pg/mL, median (IQR)	0.1 (0.0–0.6)	0.1 (0.0–0.5)	0.892
IL-2, pg/mL, median (IQR)	2.5 (1.9–3.1)	2.9 (2.2–3.5)	0.202
IL-4, pg/mL, median (IQR)	0.5 (0.3–1.4)	0.4 (0.3–1.3)	0.849
IL-5, pg/mL, median (IQR)	3.4 (2.1–5.7)	3.5 (2.4–5.1)	0.507
IL-6, pg/mL, median (IQR)	3.6 (1.0–22.9)	1.4 (0.3–9.1)	0.048
IL-7, pg/mL, median (IQR)	0.2 (0.1–0.3)	0.1 (0.1–0.2)	0.654
IL-8 (CXCL8), pg/mL, median (IQR)	0.5 (0.3–2.1)	0.3 (0.1–0.9)	0.022
IL-9, pg/mL, median (IQR)	0.1 (0.0–1.1)	0.0 (0.0–0.4)	0.078
IL-10, pg/mL, median (IQR)	0.4 (0.2–0.8)	0.3 (0.2–0.5)	0.526
IL-12 p70, pg/mL, median (IQR)	0.1 (0.1–0.3)	0.1 (0.1–0.2)	0.671
IL-13, pg/mL, median (IQR)	7.9 (2.2–20.4)	2.9 (1.1–7.0)	0.469
IL-15, pg/mL, median (IQR)	7.4 (1.3–21.9)	4.6 (0.8–18.6)	0.626
IL-17 (ACTLA8), pg/mL, median (IQR)	1.5 (0.5–6.3)	1.3 (0.7–2.9)	0.469
IL-18, pg/mL, median (IQR)	7.3 (5.2–15.0)	5.4 (4.0–8.6)	0.040
IL-21, pg/mL, median (IQR)	61.2 (4.7–246.3)	23.4 (6.3–162.2)	0.761
IL-22, pg/mL, median (IQR)	74.9 (23.5–188.0)	47.6 (7.7–227.9)	0.593
IL-23, pg/mL, median (IQR)	0.01 (0.0–0.02)	0.0 (0.0–0.01)	0.046
IL-27, pg/mL, median (IQR)	3.4 (0.8–24.8)	2.8 (0.8–8.0)	0.378
IP10 (CXCL10), pg/mL, median (IQR)	9.3 (5.7–14.4)	7.0 (4.5–12.2)	0.175
MCP-1/CCL2, pg/mL, median (IQR)	23.6 (15.8–36.2)	15.3 (5.9–26.9)	0.021
MIP1 alpha (CCL3), pg/mL, median (IQR)	0.5 (0.1–1.9)	0.3 (0.1–0.9)	0.370
MIP1 beta (CCL4), pg/mL, median (IQR)	14.7 (9.8–50.2)	10.1 (5.5–21.6)	0.043
RANTES (CCL5), pg/mL, median (IQR)	16.7 (15.2–20.6)	16.9 (14.4–20.6)	0.594
SDF-1 alpha, pg/mL, median (IQR)	669.0 (405.2–1333.0)	389.9 (278.0–630.4)	0.033
TNF alpha, pg/mL, median (IQR)	1.1 (0.6–3.0)	0.9 (0.7–1.5)	0.597
TNF beta, pg/mL, median (IQR)	0.0 (0.0–0.1)	0.0 (0.0–0.0)	0.431

Abbreviations: RA: rheumatoid arthritis; ILD: interstitial lung disease; DAS28-ESR: 28-joint Disease Activity Score with erythrocyte sedimentation rate; HAQ: Health Assessment Questionnaire; GM-CSF: granulocyte macrophage-colony stimulating factor; IFN: interferon; IL: interleukin; TNF: tumor necrosis factor; IP10 (CXCL10): C-X-C motif chemokine ligand 10; RANTES (CCL5): chemokine ligand 5; SDF-1 alpha: stromal cell-derived factor 1.

**Table 3 ijms-24-06800-t003:** Cox regression model of factors associated with ILD in patients with RA.

Dependent Variable	Predictor	HR	95% CI	*p* Value
ILD				
	DAS28-ESR, moderate-high activity	2.474	1.173–5.220	0.017
	High ACPA titer (>340)	2.905	1.244–6.786	0.014
	IL-18, pg/mL	1.063	1.002–1.127	0.044
	MCP-1/CCL2, pg/mL	1.031	1.001–1.064	0.049
	SDF-1 alpha, pg/mL	1.001	1.001–1.002	0.010

Abbreviations: RA: rheumatoid arthritis; ILD: interstitial lung disease; DAS28-ESR: 28-joint Disease Activity Score with erythrocyte sedimentation rate; ACPA: anti–citrullinated peptide antibody; IL: interleukin. Variables included in the equation: age, sex, DAS28-ESR, high ACPA titer, IL-1 alpha, IL-18, IL-6, IL8 (CXCL8), MCP-1/CCL2, MCP-1/CCL4, SDF-1 alpha.

**Table 4 ijms-24-06800-t004:** Inflammatory activity and cytokine profile in 35 patients with RA-ILD according to progression of lung disease.

Variable	RA with Progression of ILD N = 13	RA without Progression of ILD N = 22	*p* Value
Activity and physical function indices			
DAS28-ESR, mean (SD)	3.2 (1.0)	3.1 (0.9)	0.602
Remission-low disease activity, n (%)	6 (46.2)	13 (59.1)	0.347
Moderate-high disease activity, n (%)	7 (53.8)	9 (40.9)	0.347
HAQ, mean (SD)	1.3 (0.7)	1.2 (0.6)	0.511
Cytokines			
Eotaxin CCL11, pg/mL, median (IQR)	11.6 (7.2–19.7)	14.6 (6.1–20.2)	0.176
GM-CSF, pg/mL, median (IQR)	7.5 (3.4–15.2)	5.0 (3.1–8.0)	0.232
GRO alpha (CXCL1), pg/mL, median (IQR)	0.3 (0.2–0.4)	0.3 (0.2–0.3)	0.552
IFN gamma, pg/mL, median (IQR)	0.8 (0.5–1.6)	0.9 (0.6–1.6)	0.753
IL-1 alpha, pg/mL, median (IQR)	4.9 (1.0–10.3)	0.2 (0.2–1.5)	0.020
IL-1 beta, pg/mL, median (IQR)	0.1 (0.0–0.8)	0.1 (0.0–0.3)	0.600
IL-2, pg/mL, median (IQR)	2.8 (2.2–3.4)	2.3 (1.9–3.0)	0.362
IL-4, pg/mL, median (IQR)	0.5 (0.2–1.5)	0.5 (0.3–1.4)	0.780
IL-5, pg/mL, median (IQR)	3.7 (2.8–6.7)	2.8 (2.1–5.2)	0.261
IL-6, pg/mL, median (IQR)	3.6 (0.7–45.4)	3.5 (0.9–18.4)	0.701
IL-7, pg/mL, median (IQR)	0.2 (0.1–0.3)	0.1 (0.1–0.3)	0.193
IL8 (CXCL8), pg/mL, median (IQR)	0.6 (0.3–3.3)	0.5 (0.3–1.8)	0.309
IL-9, pg/mL, median (IQR)	0.3 (0.0–2.1)	0.0 (0.0–0.6)	0.112
IL-10, pg/mL, median (IQR)	0.4 (0.2–2.2)	0.4 (0.2–0.7)	0.649
IL-12 p70, pg/mL, median (IQR)	0.2 (0.1–0.3)	0.1 (0.1–0.2)	0.232
IL-13, pg/mL, median (IQR)	20.1 (9.4–20.9)	2.8 (2.1–14.2)	0.267
IL-15, pg/mL, median (IQR)	11.6 (6.3–44.4)	4.9 (0.9–21.0)	0.107
IL-17 (ACTLA8), pg/mL, median (IQR)	3.4 (1.1–13.3)	0.9 (0.5–4.2)	0.148
IL-18, pg/mL, median (IQR)	17.0 (8.6–19.1)	6.0 (5.1–9.3)	0.005
IL-21, pg/mL, median (IQR)	149.3 (30.8–313.7)	9.2 (2.9–119.0)	0.073
IL-22, pg/mL, median (IQR)	129.5 (41.5–252.4)	37.1 (20.1–148.1)	0.270
IL-23, pg/mL, median (IQR)	0.0 (0.0–0.0)	0.0 (0.0–0.0)	0.753
IL-27, pg/mL, median (IQR)	2.3 (0.8–98.5)	4.5 (0.8–27.3)	0.876
IP10 (CXCL10), pg/mL, median (IQR)	11.0 (7.2–18.8)	8.3 (5.2–11.3)	0.193
MCP-1/CCL2, pg/mL, median (IQR)	23.9 (14.5–40.2)	11.8 (8.0–23.6)	0.041
MIP1 alpha CCL3, pg/mL, median (IQR)	1.6 (0.2–5.2)	0.3 (0.1–0.7)	0.060
MIP1 beta CCL4, pg/mL, median (IQR)	47.9 (13.9–81.7)	23.4 (16.1–33.8)	0.701
RANTES (CCL5), pg/mL, median (IQR)	16.7 (15.7–19.4)	17.9 (14.5–21.4)	0.861
SDF-1 alpha, pg/mL, median (IQR)	1300.7 (528.9–2837.6)	523.9 (357.6–747.5)	0.050
TNF alpha, pg/mL, median (IQR)	1.1 (0.5–3.2)	1.1 (0.6–1.6)	0.889
TNF beta, pg/mL, median (IQR)	0.0 (0.0–0.3)	0.0 (0.0–0.1)	0.990

Abbreviations: RA: rheumatoid arthritis; ILD: interstitial lung disease; DAS28-ESR: 28-joint Disease Activity Score with erythrocyte sedimentation rate; HAQ: Health Assessment Questionnaire; GM-CSF: granulocyte macrophage-colony stimulating factor online; IFN: interferon; IL: interleukin; TNF: tumor necrosis factor; IP10 (CXCL10): C-X-C motif chemokine ligand 10; RANTES (CCL5): chemokine ligand 5; SDF-1 alpha: stromal cell-derived factor 1.

## Data Availability

The datasets used and/or analyzed in the present study are available from the corresponding author upon reasonable request.

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
