# Peer review of "Inflammatory Biomarkers in the Diagnosis and Prognosis of Rheumatoid Arthritis–Associated Interstitial Lung Disease"

_ijms, 2023, doi:10.3390/ijms24076800_

Round 1
Reviewer 1 Report
What is the cut-off point for IL18 levels to allow obtaining a HR underestimated the value of HR??
Lines 37 and 38 MCP-1/CCL2 (HR [95% CI], 1.03 37 [1.00-1.06]; p=0.049), and SDF-1 (HR [95% CI], 1.00 [1.00-1.00], p = 0.010). The only associated variable.
Materials and methods
How many patients >16 years???
The 16-year-old patients included in the study have probably juvenile rheumatoid arthritis?
Lines 104-105 and treatment with disease-modifying antirheumatic drugs (DMARDs). The patients with ILD were found with antifibrosant treatment in the case of the px with UIP pattern.
Despite showing no difference between the comparison groups, 37.1% of ILD patients received abatacept, on the other hand, 31.4% of RA-only patients received anti-TNF, which could be reaching levels of proteins to be determined.
Lines 111-112 Each case was matched with a control 111 by sex, age (±3 years), and time since diagnosis (±24 months). In one year of treatment, even the possible markers they propose could be modified, despite performing a paired analysis, was it adjusted for these covariates?
A group of individuals without the diseases would not be expected, that is, without RA and without ILD allow differentiating both diseases that can generate systemic inflammation.
Lines 119-120 A blood sample was taken to 119 measure levels of cytokines and other inflammatory parameters. All patients with RA
When was the sample taken?, was the basal to the dx of AR or ILD?
Lines 128 y 129 The 2 main outcomes were (1) presence of RA-ILD and (2) progression of lung disease in patients with RA-ILD. RA-ILD was confirmed by the presence of respiratory symp...
How were these patients who had an outcome with the presence of RA-ILD considered? In which study group is included???
Lines 153-163 What are the detection limits of these kits for the determination of multiplex proteins, since in the case of IL18 referred to in the ProcartaPlexHuman Th1/Th2 Cytokine Panel 11plex kit, they range from 64100/16 pg/mL (ULOQ/LLOQ), and the results obtained that are shown in table 2 (RA-ILD=7.3 pg/mL and RA no ILD= 5.4 pg/mL) are below limits.
In Table 1, the name of the control group (RA without ILD), I recommend unifying the name of this group, since there are: RA without ILD, RA-ILD control, and RA no ILD.
- Time with ILD, months, mean (SD) 66.1 (47.2), aquí se incluyen los pacientes que en el seguimiento presentaron ILD???
- High ACPA titer (>340)
In methodology, levels >20 IU/mL that was considered positive is mentioned, but this cut-off point is not mentioned, why >340?
Figure 1 is not completely clear, due to the nomenclatures for the group of individuals without ILD, for which reason RA-ILD controls selected 2021 (n=38) could be modified as shown in the RA without ILD tables, in all cases mentioned only RA.
RA-ILD cases presented greater disease activity, greater joint inflammation, and lower physical function (HAQ), in the case of cytokines.
In Table 2, the following variables
|
Variable |
RA-ILD N = 35 |
RA without ILD N = 35 |
p Value |
|
DAS28-ESR, mean (SD) |
3.1 (0.9)
|
2.6 (0.9)
|
0.032 |
|
Number of painful joints, median (IQR) |
0.0 (0.0-1.0) |
0.0 (0.0-1.0) |
0.792 |
|
Number of swollen joints, median (IQR) |
0.0 (0.0-1.0) |
0.0 (0.0-0.0) |
0.040 |
One suggestion is that it be carried out only in those who show moderate-high activity (the 16 patients with RA-ILD and the 8 RA patients without ILD) since all patients who do not have activity are being considered and those who are not considered. they do have inflammation and joint pain or exclude patients with disease remission-low activity.
In table 3, why not include the treatment? or at least the drugs that had significant differences shown in Table 1 Adherence to treatment between biological drugs and synthetic conventional ones
How were the covariates included within the predictors?
There could be a bias within the analysis carried out to determine the HR of each predictor.
what is the cut-off point used in IL18, MCP-1, and SDF-1?
How does each of these variables affect the predictors, do they all participate in the same way? despite the differences between the groups.
Line 251-252 The titles are not presented, they only show the ACPA cut-offs, that is, they show how many patients were >20 and >340, however, how much is the mean/median of ACPA detected???
Line 286-287 The sensitivity and specificity of ACPA for RA are high, however, only these have the same predictive capacity in ILD?
Author Response
Comments for the reviewers
We would like to thank the editor for considering our work for publication in “International Journal of Molecular Sciences” and the reviewers for their comments, which have helped to improve the quality of our manuscript.
Below, we provide a point-by-point reply to the comments.
Reviewer #1:
- What is the cut-off point for IL18 levels to allow obtaining a HR underestimated the value of HR??
Reply: The authors apologies about the lack of the complete information regarding this variable. It should be noted that IL-18 does not have a recognizable cut-off point and the variable was evaluated quantitatively in pg/mL. To address this issue, we have made sure to include these units throughout the document, including in the abstract.
-Page 1; line 45: “IL-18 in pg/mL (HR [95% CI], 1.06 [1.00-1.12]; p=0.044),”
-Page 7; lines 253-256: “The variables that were independently associated with ILD were moderate-high disease activity according to the DAS28-ESR (β= 0.914; p=0.017), and high levels of ACPA (β= 1.064; p=0.014), IL-18 in pg/mL (β= 0.091; p=0.044), MCP-1/CCL2 in pg/mL (β= 0.031; p=0.049), and SDF-1 alpha in pg/mL (β= 0.081; p=0.010).”
-Page 9; lines 289-291: “The only variable that was associated with progression of RA-ILD in this model was IL-18 in pg/mL (β= 0.227; p=0.004) (Figure 2).”
- Lines 37 and 38 MCP-1/CCL2 (HR [95% CI], 1.03 37 [1.00-1.06]; p=0.049), and SDF-1 (HR [95% CI], 1.00 [1.00-1.00], p = 0.010). The only associated variable.
Reply: The authors apologies about the lack of the complete information regarding this variable. We have added these units throughout the document, including in the abstract.
-Page 1; lines 43-48: “The variables associated with RA-ILD in the Cox regression analysis were disease activity (DAS28) (HR [95% CI], 2.47 [1.17-5.22]; p=0.017) and high levels of ACPA (HR [95% CI], 2.90 [1.24-6.78]; p=0.014), IL-18 in pg/mL (HR [95% CI], 1.06 [1.00-1.12]; p=0.044), MCP-1/CCL2 in pg/mL (HR [95% CI], 1.03 [1.00-1.06]; p=0.049), and SDF-1 in pg/mL (HR [95% CI], 1.00 [1.00-1.00]; p=0.010). The only variable associated with progression of ILD was IL-18 in pg/mL (HR [95% CI], 1.25 [1.07-1.46]; p=0.004).”
-Page 7; lines 253-256: “The variables that were independently associated with ILD were moderate-high disease activity according to the DAS28-ESR (β= 0.914; p=0.017), and high levels of ACPA (β= 1.064; p=0.014), IL-18 in pg/mL (β= 0.091; p=0.044), MCP-1/CCL2 in pg/mL (β= 0.031; p=0.049), and SDF-1 alpha in pg/mL (β= 0.081; p=0.010).”
- Materials and methods. How many patients >16 years??? The 16-year-old patients included in the study have probably juvenile rheumatoid arthritis?
Reply: The study included 70 patients diagnosed with rheumatoid arthritis who were over 16 years old. Details regarding the inclusion criteria can be found in the material and method of work of the paper:
-Page 3; lines 116-118: “2.1 Study design and participants: The inclusion criteria were age ≥16 years, a diagnosis of RA based on the 2010 classification criteria of the American College of Rheumatology/European League Against Rheumatism (ACR/EULAR 2010) (22),”
- Lines 104-105 and treatment with disease-modifying antirheumatic drugs (DMARDs). The patients with ILD were found with antifibrosant treatment in the case of the px with UIP pattern.
Reply: We appreciate your comment. Only one patient was taking antifibrotic medication combined with mycophenolate mofetil and abatacept. This patient had ILD with UIP pattern. The treatments of the included patients are shown in the results section. The antifibrotic treatment has been added in Table 1.
-Page 5; lines 219-224: “All the participants were receiving a DMARD (Table 1). While there were no differences in terms of the percentage of csDMARDs and bDMARDs, the cases less frequently took methotrexate (p=0.040) and anti-TNF agents (p=0.041) and more frequently took hydroxychloroquine (p=0.010) and abatacept (p=0.004). Only 4 patients with RA-ILD were receiving mycophenolate mofetil (p=0.032), and 1 patient was taking antifibrotic medication combined with mycophenolate mofetil and abatacept. “
Table 1: Baseline characteristics of the study population
|
VARIABLE |
RA-ILD N = 35 |
RA without ILD N = 35 |
p Value |
|
Epidemiological characteristics |
|
|
|
|
Age, years, mean (SD) |
69.7 (9.3) |
66.6 (7.0) |
0.130 |
|
Male sex; n (%) |
20 (57.1) |
20 (57.1) |
1.000 |
|
Clinical-analytical characteristics |
|
|
|
|
Smoking history |
|
|
0.760 |
|
Nonsmokers, n (%) |
17 (48.6) |
18 (51.4) |
|
|
Ex-smokers, n (%) |
10 (28.6) |
8 (22.9) |
|
|
Active smokers, n (%) |
8 (22.9) |
9 (25.7) |
|
|
Time with RA, months, median (IQR) |
149.8 (93.3-245.5) |
133.7 (67.8-204.2) |
0.384 |
|
Time with ILD, months, mean (SD) |
66.1 (47.2) |
- |
- |
|
RF+ (>10), n (%) |
33 (94.3) |
31 (88.6) |
0.393 |
|
High RF (>60), n (%) |
24 (68.6) |
17 (48.6) |
0.089 |
|
ACPA+ (>20), n (%) |
32 (91.4) |
31 (88.6) |
0.690 |
|
ACPA titer, median (IQR) |
562.0 (150.0-1305.0) |
283.0 (41.0-816.0) |
0.073 |
|
High ACPA titer (>340), n (%) |
22 (63.0) |
14 (40.0) |
0.039 |
|
Radiographic erosions, n (%) |
21 (60.0) |
19 (55.6) |
0.705 |
|
Current treatment |
|
|
|
|
csDMARD, n (%) |
28 (80.0) |
33 (94.3) |
0.074 |
|
Methotrexate, n (%) |
19 (54.3) |
27 (77.1) |
0.040 |
|
Leflunomide, n (%) |
3 (8.6) |
6 (17.1) |
0.284 |
|
Sulfasalazine, n (%) |
2 (5.7) |
2 (5.7) |
1.000 |
|
Hydroxychloroquine, n (%) |
6 (17.1) |
0 (0.0) |
0.010 |
|
Mycophenolate, n (%) |
4 (11.4) |
0 (0.0) |
0.032 |
|
bDMARD, n (%) |
22 (62.9) |
18 (51.4) |
0.334 |
|
Anti-TNF, n (%) |
4 (11.4) |
11 (31.4) |
0.041 |
|
Tocilizumab, n (%) |
3 (8.6) |
2 (5.7) |
0.643 |
|
Abatacept, n (%) |
13 (37.1) |
3 (8.6) |
0.004 |
|
Rituximab, n (%) |
2 (5.7) |
0 (0.0) |
0.151 |
|
JAK inhibitor, n (%) |
0 (0.0) |
2 (5.7) |
0.151 |
|
Corticosteroids, n (%) |
22 (62.9) |
6 (17.1) |
0.001 |
|
Nintedanib, n (%) |
1 (2.9) |
0 (0.0) |
0.314 |
|
Dose of corticosteroid, mg/d, mean (SD) |
7.9 (4.0) |
5.0 (0.0) |
0.135 |
|
Pulmonary function tests |
|
|
|
|
FVC <80%, n (%) |
28 (80.0) |
5 (14.3) |
<0.001 |
|
FVC predicted (%), mean (SD) |
63.0 (17.1) |
83.4 (4.4) |
<0.001 |
|
FEV1 <80%, n (%) |
23 (67.6) |
5 (14.3) |
0.001 |
|
FEV1 predicted (%), mean (SD) |
68.7 (15.9) |
84.0 (11.5) |
<0.001 |
|
DLCO <80%, n (%) |
29 (85.3) |
3 (8.6) |
<0.001 |
|
DLCO-SB predicted (%), mean (SD) |
61.0 (15.2) |
85.9 (7.9) |
<0.001 |
|
HRCT pattern |
|
|
|
|
UIP, n (%) |
29 (82.9) |
0 (0.0) |
<0.001 |
|
NSIP, n (%) |
6 (17.1) |
0 (0.0) |
<0.001 |
Abbreviations. RA: rheumatoid arthritis; ILD: interstitial lung disease; SD: standard deviation; RF: rheumatoid factor; ACPA: anti–citrullinated peptide antibody; csDMARD: conventional synthetic disease-modifying antirheumatic drug; bDMARD: biologic disease-modifying antirheumatic drug; FVC: forced vital capacity; FEV1: forced expiratory volume in the first second; DLCO: diffusing capacity of the lung for carbon monoxide; HRCT: high-resolution computed tomography; UIP: usual interstitial pneumonia; NSIP: nonspecific interstitial pneumonia.
- Despite showing no difference between the comparison groups, 37.1% of ILD patients received abatacept, on the other hand, 31.4% of RA-only patients received anti-TNF, which could be reaching levels of proteins to be determined.
Reply: As per the reviewer's comments, RA patients with ILD were found to have a higher frequency of abatacept usage, whereas RA patients without ILD more frequently had an anti-TNF. It is important to note that these treatments may directly impact cytokines and alter the inflammatory process, thereby affecting the results. However, this is an observational study of clinical practice, it should be noted that treatments are the most common used therapies in both groups of patients. In our clinical practice, we tend to avoid prescribing anti-TNF drugs to patients with ILD and instead prefer the use of abatacept. This comment has also highlighted the limitations of our study.
-Page 12; lines 368-370: “It is also noteworthy that all the participants had received DMARDs and that these drugs directly affect cytokines and re-establish the inflammatory process. Nevertheless, since both the cases and the controls were treated with DMARDs, this effect applied to all the participants included.”
- Lines 111-112 Each case was matched with a control 111 by sex, age (±3 years), and time since diagnosis (±24 months). In one year of treatment, even the possible markers they propose could be modified, despite performing a paired analysis, was it adjusted for these covariates?
Reply: In the inclusion criteria, we have specified that all patients had been undergoing treatment with stable DMARD for at least a year. This measure was taken to prevent, as much as possible, any modifications due to treatment changes. It is important to note that, as mentioned earlier, these treatments may directly affect cytokines and alter the inflammatory process, although this would apply equally to all patients. This information has been detailed in the material and methods section of our study, and we have also included it in the study limitations.
-Page 3; lines 116-120: “The inclusion criteria were age ≥16 years, a diagnosis of RA based on the 2010 classification criteria of the American College of Rheumatology/European League Against Rheumatism (ACR/EULAR 2010) (22), ILD diagnosed according to the criteria of the American Thoracic Society/European Respiratory Society (23), and treatment with disease-modifying antirheumatic drugs (DMARDs) for at least one year.”
-Page 12; lines 369-372: “It is also noteworthy that all the participants had received DMARDs and that these drugs directly affect cytokines and re-establish the inflammatory process. Nevertheless, since both the cases and the controls were treated with DMARDs, this effect applied to all the participants included.”
- A group of individuals without the diseases would not be expected, that is, without RA and without ILD allow differentiating both diseases that can generate systemic inflammation.
Reply: We appreciate the reviewer's comment. The aim of our study was to identify soluble cytokine biomarkers in patients with RA-ILD and to describe their association with other factors involved in the progression of lung disease. Consequently, we believe that including a healthy control group was not essential for the purposes of this study. Additionally, performing HCRT in healthy individuals may raise ethical concerns. Nevertheless, we have taken note of the reviewer's recommendation and included this comment in the discussion section of our study.
-Page 12; lines 371-374: “Furthermore, we have not included a healthy control group without RA, since the main objective was to identify soluble cytokine biomarkers in patients with RA-ILD and progression of lung disease, and performing HCRT in healthy controls would be more controversial.”
- Lines 119-120 A blood sample was taken to 119 measure levels of cytokines and other inflammatory parameters. All patients with RA. When was the sample taken?, was the basal to the dx of AR or ILD?
Reply: We appreciate your comment. A blood sample was taken to measure cytokine levels and other inflammatory parameters from all patients at inclusion. The Time with RA and ILD, in months, at inclusion in the study is shown in Table 1.
-Page 3; lines 135-136: “A blood sample was taken to measure levels of cytokines and other inflammatory parameters from all patients at inclusion.”
- Lines 128 y 129 The 2 main outcomes were (1) presence of RA-ILD and (2) progression of lung disease in patients with RA-ILD. RA-ILD was confirmed by the presence of respiratory symp... How were these patients who had an outcome with the presence of RA-ILD considered? In which study group is included???
Reply: As per the reviewer's comments, the presence of respiratory symptoms such as dyspnea and/or cough, any identifiable ILD on the HRCT scan, and/or lung biopsy were used to confirm RA-ILD. Additionally, ILD was classified based on the standard criteria of the American Thoracic Society/European Respiratory Society International Multidisciplinary Consensus Classification of the Idiopathic Interstitial Pneumonias, which recognizes three patterns: nonspecific interstitial pneumonia (NSIP), usual interstitial pneumonia (UIP), and other patterns such as bronchiolitis obliterans, organizing pneumonia, lymphoid interstitial pneumonia, and mixed patterns (32).
Regarding the progression of lung disease: we defined three stages: (1) progression (worsening of forced vital capacity [FVC] >10% or the diffusing capacity of the lungs for carbon monoxide [DLCO] >15%); (2) nonprogression (stability or improvement in FVC ≤10% or DLCO ≤15%); and (3) improvement (increase in FVC >10% or DLCO >15%) (11). Furthermore, all patients with RA without ILD underwent HRCT and PFR at inclusion to confirm the absence of lung involvement.
-Page 3; lines 145-156: “The 2 main outcomes were (1) presence of RA-ILD and (2) progression of lung disease in patients with RA-ILD. RA-ILD was confirmed by the presence of respiratory symptoms, any identifiable ILD on the HRCT scan, and/or lung biopsy. ILD was classified according to the standard criteria of the American Thoracic Society/European Respiratory Society International Multidisciplinary Consensus Classification of the Idiopathic Interstitial Pneumonias, which recognizes 3 patterns: nonspecific interstitial pneumonia (NSIP), usual interstitial pneumonia (UIP), and other patterns (bronchiolitis obliterans, organizing pneumonia, lymphoid interstitial pneumonia, and mixed patterns) (32).
As for progression of lung disease, we defined 3 stages: (1) progression (worsening of forced vital capacity [FVC] >10% or the diffusing capacity of the lungs for carbon monoxide [DLCO] >15%); (2) nonprogression (stability or improvement in FVC ≤10% or DLCO ≤15%); and (3) improvement (increase in FVC >10% or DLCO >15%) (11).”
- Lines 153-163 What are the detection limits of these kits for the determination of multiplex proteins, since in the case of IL18 referred to in the ProcartaPlexHuman Th1/Th2 Cytokine Panel 11plex kit, they range from 64100/16 pg/mL (ULOQ/LLOQ), and the results obtained that are shown in table 2 (RA-ILD=7.3 pg/mL and RA no ILD= 5.4 pg/mL) are below limits.
Reply: We fully concur with the reviewer's comment regarding the LLOQ values of the 34-Plex kit used in our study. While it is true that the range of many measured cytokines falls below the LLOQ value, we believe that the LLOQ values for this kit are excessively restrictive. For instance, we have previously calculated the limit of detection (LOD) according to the ICH Q2(R2) Validation of analytical procedures - Scientific guideline from the European Medicine Agency (https://www.ema.europa.eu/en/ich-q2r2-validation-analytical-procedures-scientific-guideline) for the most significant cytokines in this study, namely IL-18, MCP-1, and SDF-1. The guidelines specify the calculation of LOD as 3.3*σ/s, where "σ" represents the standard deviation of the blanks and "s" represents the slope of the standard curve. Based on this, we determined LOD values of 1.26, 0.65, and 15.24 pg/mL for IL-18, MCP-1, and SDF-1, respectively. These LOD values fall below the lowest concentration values we managed to detect in our study. Moreover, the Variation Coefficients of the blanks and lowest calibrators in our assay were all close to, or below, the desirable 12%, indicating acceptable accuracy of the measurements. Therefore, we believe that our findings are reliable and valid despite the LLOQ limitations of the 34-Plex kit.
- In Table 1, the name of the control group (RA without ILD), I recommend unifying the name of this group, since there are: RA without ILD, RA-ILD control, and RA no ILD.
- Time with ILD, months, mean (SD) 66.1 (47.2), aquí se incluyen los pacientes que en el seguimiento presentaron ILD???
- High ACPA titer (>340). In methodology, levels >20 IU/mL that was considered positive is mentioned, but this cut-off point is not mentioned, why >340?
Reply: We are grateful to the reviewer for providing us with a valuable critique, and we apologize for any confusion caused by using different terminology in the manuscript, tables, and figure. We have now unified the term throughout the document. In fact, the patients with RA-ILD included in our study had a mean time with RA of 149.8 months. The time with ILD for these patients was 66.1 (47.2) months on average. We defined ACPA positivity as greater than 20 IU/mL, and high ACPA titer as greater than 340 IU/mL. We chose this cut-off point because some laboratory kits are unable to detect ACPA levels higher than 340 IU/mL, and therefore consider this value as a high titer.
-Page 4; lines 182-184: Rheumatoid factor (RF) was considered positive if >10 IU/mL, ACPA was considered positive if >20 IU/mL, and high ACPA titer if >340 IU/mL.
- Figure 1 is not completely clear, due to the nomenclatures for the group of individuals without ILD, for which reason RA-ILD controls selected 2021 (n=38) could be modified as shown in the RA without ILD tables, in all cases mentioned only RA.
Reply: According to the reviewer, we have unified the nomenclature of the groups throughout the manuscript, tables, and figure.
Figure 1: Flow diagram of patient selection into the study
- RA-ILD cases presented greater disease activity, greater joint inflammation, and lower physical function (HAQ), in the case of cytokines. In Table 2, the following variables. One suggestion is that it be carried out only in those who show moderate-high activity (the 16 patients with RA-ILD and the 8 RA patients without ILD) since all patients who do not have activity are being considered and those who are not considered. they do have inflammation and joint pain or exclude patients with disease remission-low activity.
|
Variable |
RA-ILD N = 35 |
RA without ILD N = 35 |
p Value |
|
DAS28-ESR, mean (SD) |
3.1 (0.9)
|
2.6 (0.9)
|
0.032 |
|
Number of painful joints, median (IQR) |
0.0 (0.0-1.0) |
0.0 (0.0-1.0) |
0.792 |
|
Number of swollen joints, median (IQR) |
0.0 (0.0-1.0) |
0.0 (0.0-0.0) |
0.040 |
Reply: In response to the reviewer's recommendations, we have limited the cytokine comparison to only those patients who exhibited moderate to high disease activity, including the 16 patients with RA-ILD and 8 patients with RA without ILD. Our results showed that patients with RA-ILD had higher levels of IL-18 pg/mL (7.2 [4.9-14.7] vs 5.7 [3.5-10.4]; p=0.045), SDF-1 alpha pg/mL (885.7 [523.9-1905.7] vs 574.0 [229.8-890.6]; p=0.096), IL-8 (CXCL8) pg/mL (0.6 [0.4-3.5] vs 0.3 [0.2-1.3]; p=0.061), and IL-1 pg/mL (1.7 [0.3-8.2] vs 0.6 [0.1-4.4]; p=0.130) compared to RA patients without ILD. Due to the small sample size, statistical significance was not reached in some of the cytokines. However, we have mentioned in the text that in patients with moderate to high inflammatory activity, patients with RA-ILD exhibited higher levels of IL-18 pg/mL. Furthermore, we have included a supplementary table containing all the cytokine data..
Page 7; lines 249-250: “In patients with moderate-high inflammatory activity, RA ILD patients had higher values of IL-18 (Supplementary table 1).”
Supplementary table 1: Cytokine profile of 16 patients with RA-ILD and 8 RA patients without ILD with moderate-high activity
|
Variable |
RA-ILD N = 16 |
RA without ILD N = 8 |
p Value |
|
Cytokines |
|
|
|
|
Eotaxin (CCL11), pg/mL, median (IQR) |
14.7 (11.4-26.5) |
8.9 (7.1-15.1) |
0.237 |
|
GM-CSF, pg/mL, median (IQR) |
5.5 (3.6-8.5) |
3.6 (3.0-14.7) |
0.581 |
|
GRO alpha (CXCL1), pg/mL, median (IQR) |
0.3 (0.2-0.4) |
0.3 (0.2-0.5) |
0.690 |
|
IFN gamma, pg/mL, median (IQR) |
0.8 (0.4-1.3) |
0.7 (0.5-1.3) |
0.837 |
|
IL-1 alpha, pg/mL, median (IQR) |
1.7 (0.3-8.2) |
0.6 (0.1-4.4) |
0.130 |
|
IL-1 beta, pg/mL, median (IQR) |
0.1 (0.0-0.3) |
0.1 (0.0-1.0) |
0.731 |
|
IL-2, pg/mL, median (IQR) |
2.4 (2.1-3.6) |
2.5 (2.2-4.5) |
0.630 |
|
IL-4, pg/mL, median (IQR) |
0.6 (0.3-1.1) |
0.8 (0.2-2.1) |
0.600 |
|
IL-5, pg/mL, median (IQR) |
3.4 (2.8-6.7) |
4.1 (2.1-7.5) |
0.783 |
|
IL-6, pg/mL, median (IQR) |
3.6 (1.4-21.7) |
1.7 (0.1-16.5) |
0.298 |
|
IL-7, pg/mL, median (IQR) |
0.2 (0.1-0.2) |
0.1 (0.1-0.4) |
0.783 |
|
IL-8 (CXCL8), pg/mL, median (IQR) |
0.6 (0.4-3.5) |
0.3 (0.2-1.3) |
0.061 |
|
IL-9, pg/mL, median (IQR) |
0.2 (0.0-1.7) |
0.0 (0.0-0.7) |
0.123 |
|
IL-10, pg/mL, median (IQR) |
0.4 (0.1-0.8) |
0.3 (0.3-1.1) |
0.630 |
|
IL-12 p70, pg/mL, median (IQR) |
0.1 (0.1-0.2) |
0.1 (0.1-0.4) |
0.890 |
|
IL-13, pg/mL, median (IQR) |
4.9 (2.1-10.0) |
1.1 (0.7-4.1) |
0.500 |
|
IL-15, pg/mL, median (IQR) |
7.6 (1.5-35.8) |
4.1 (0.8-18.6) |
0.445 |
|
IL-17 (ACTLA8), pg/mL, median (IQR) |
1.5 (0.5-6.3) |
1.7 (0.2-11.2) |
0.630 |
|
IL-18, pg/mL, median (IQR) |
7.2 (4.9-14.7) |
5.7 (3.5-10.4) |
0.045 |
|
IL-21, pg/mL, median (IQR) |
94.7 (13.2-277.8) |
84.4 (18.2-607.1) |
0.622 |
|
IL-22, pg/mL, median (IQR) |
85.0 (19.7-161.7) |
67.8 (12.9-147.8) |
0.624 |
|
IL-23, pg/mL, median (IQR) |
0.01 (0.0-0.01) |
0.0 (0.0-0.01) |
0.267 |
|
IL-27, pg/mL, median (IQR) |
3.2 (1.0-28.0) |
2.9 (0.8-9.5) |
0.875 |
|
IP10 (CXCL10), pg/mL, median (IQR) |
10.8 (6.7-17.3) |
9.0 (6.3-14.3) |
0.731 |
|
MCP-1/CCL2, pg/mL, median (IQR) |
25.0 (21.4-36.5) |
18.5 (9.1-30.7) |
0.170 |
|
MIP1 alpha (CCL3), pg/mL, median (IQR) |
0.9 (0.3-4.4) |
0.4 (0.2-5.0) |
0.490 |
|
MIP1 beta (CCL4), pg/mL, median (IQR) |
27.5 (13.3-56.2) |
16.4 (5.9-50.0) |
0.132 |
|
RANTES (CCL5), pg/mL, median (IQR) |
16.3 (14.3-21.9) |
16.5 (13.8-218.0) |
0.535 |
|
SDF-1 alpha, pg/mL, median (IQR) |
885.7 (523.9-1905.7) |
574.0 (229.8-890.0) |
0.096 |
|
TNF alpha, pg/mL, median (IQR) |
1.1 (0.6-2.2) |
0.8 (0.5-2.8) |
0.407 |
|
TNF beta, pg/mL, median (IQR) |
0.0 (0.0-0.1) |
0.0 (0.0-0.0) |
0.368 |
Abbreviations; RA: rheumatoid arthritis; ILD: interstitial lung disease; GM-CSF: granulocyte macrophage-colony stimulating factor; IFN: interferon; IL: interleukin; TNF: tumor necrosis factor; IP10 (CXCL10): C-X-C motif chemokine ligand 10; RANTES (CCL5): chemokine ligand 5; SDF-1 alpha: stromal cell-derived factor 1.
- In table 3, why not include the treatment? or at least the drugs that had significant differences shown in Table 1 Adherence to treatment between biological drugs and synthetic conventional ones
Reply: We have attempted to apply alternative multivariate models that take into account the treatments, but we did not observe any changes in the models.
- How were the covariates included within the predictors?. There could be a bias within the analysis carried out to determine the HR of each predictor.
Reply: We have used stepwise multiple Cox regression analysis (Wald). The factors that we have included in the model were those that were significant in the univariate analysis. The multicollinearity of the independent variables was checked using the Pearson correlation coefficient. If the r coefficient between the 2 variables was >0.4, we included them separately in the models and chose the one with the best explained the dependent variable. We have added this comment to the document.
-Page 4; lines 202-208: “Finally, we ran 2 stepwise Cox regression analysis (Wald), one to identify severity and inflammatory factors associated with RA-ILD adjusted for time since diagnosis of RA and another to determine the factors associated with progression of RA-ILD adjusted for time since diagnosis of RA-ILD. The factors that we have included in the model were those that were significant in the univariate analysis. The multicollinearity of the independent variables was checked using the Pearson correlation coefficient. “
- what is the cut-off point used in IL18, MCP-1, and SDF-1?
Reply: The authors apologies about the lack of the complete information regarding this variable. These are quantitative variables. We have added these units throughout the document, including in the abstract.
-Page 4; lines 175-181: “The analysis included cytokines levels that intervene in Th1/Th2 function (GM-CSF, IFN gamma, IL-1 beta, IL-2, IL-4, IL-5, IL-6, IL-8, IL-12p70, IL-13, IL-18, TNF alpha) and Th9/Th17/Th22/Treg function (IL-9, IL-10, IL-17A [CTLA-8], IL-21, IL-22, IL-23, IL-27), inflammatory cytokines (IFN alpha, IL-1 alpha, IL-7, IL-15, IL-31, TNF beta), and chemokines (eotaxin [CCL11], GRO alpha [CXCL1], IP-10 [CXCL10], monocyte chemoattractant protein 1 [MCP-1]/CCL2, MIP-1 alpha [CCL3], MIP-1 beta [CCL4], RANTES [CCL5], and stromal cell–derived factor 1 alpha [SDF-1 alpha]).”
- How does each of these variables affect the predictors, do they all participate in the same way? despite the differences between the groups.
Reply: Based on the reviewer's feedback, we have included the value of ß (the regression coefficient) in the results section and HR (Exp ß) in the tables and abstract. We have made every effort to provide all relevant information regarding the variables.
-Page 7; lines 253-256: “The variables that were independently associated with ILD were moderate-high disease activity according to the DAS28-ESR (β= 0.914; p=0.017), and high levels of ACPA (β= 1.064; p=0.014), IL-18 pg/mL (β= 0.091; p=0.044), MCP-1/CCL2 pg/mL (β= 0.031; p=0.049), and SDF-1 alpha pg/mL (β= 0.081; p=0.010).
-Page 9; lines 288-290: “The only variable that was associated with progression of RA-ILD in this model was IL-18 in pg/mL (β= 0.227; p=0.004) (Figure 2).”
- Line 251-252 The titles are not presented, they only show the ACPA cut-offs, that is, they show how many patients were >20 and >340, however, how much is the mean/median of ACPA detected???
Reply: We have added the value of the ACPA in table 1 "baseline characteristics of the study population"; and in Supplementary Table 3 "Baseline characteristics of 35 patients with RA-ILD and progression of lung disease".
Table 1: Baseline characteristics of the study population
|
VARIABLE |
RA-ILD N = 35 |
RA without ILD N = 35 |
p Value |
|
Epidemiological characteristics |
|
|
|
|
Age, years, mean (SD) |
69.7 (9.3) |
66.6 (7.0) |
0.130 |
|
Male sex; n (%) |
20 (57.1) |
20 (57.1) |
1.000 |
|
Clinical-analytical characteristics |
|
|
|
|
Smoking history |
|
|
0.760 |
|
Nonsmokers, n (%) |
17 (48.6) |
18 (51.4) |
|
|
Ex-smokers, n (%) |
10 (28.6) |
8 (22.9) |
|
|
Active smokers, n (%) |
8 (22.9) |
9 (25.7) |
|
|
Time with RA, months, median (IQR) |
149.8 (93.3-245.5) |
133.7 (67.8-204.2) |
0.384 |
|
Time with ILD, months, mean (SD) |
66.1 (47.2) |
- |
- |
|
RF+ (>10), n (%) |
33 (94.3) |
31 (88.6) |
0.393 |
|
High RF (>60), n (%) |
24 (68.6) |
17 (48.6) |
0.089 |
|
ACPA+ (>20), n (%) |
32 (91.4) |
31 (88.6) |
0.690 |
|
ACPA titer, median (IQR) |
562.0 (150.0-1305.0) |
283.0 (41.0-816.0) |
0.073 |
|
High ACPA titer (>340), n (%) |
22 (63.0) |
14 (40.0) |
0.039 |
|
Radiographic erosions, n (%) |
21 (60.0) |
19 (55.6) |
0.705 |
|
Current treatment |
|
|
|
|
csDMARD, n (%) |
28 (80.0) |
33 (94.3) |
0.074 |
|
Methotrexate, n (%) |
19 (54.3) |
27 (77.1) |
0.040 |
|
Leflunomide, n (%) |
3 (8.6) |
6 (17.1) |
0.284 |
|
Sulfasalazine, n (%) |
2 (5.7) |
2 (5.7) |
1.000 |
|
Hydroxychloroquine, n (%) |
6 (17.1) |
0 (0.0) |
0.010 |
|
Mycophenolate, n (%) |
4 (11.4) |
0 (0.0) |
0.032 |
|
bDMARD, n (%) |
22 (62.9) |
18 (51.4) |
0.334 |
|
Anti-TNF, n (%) |
4 (11.4) |
11 (31.4) |
0.041 |
|
Tocilizumab, n (%) |
3 (8.6) |
2 (5.7) |
0.643 |
|
Abatacept, n (%) |
13 (37.1) |
3 (8.6) |
0.004 |
|
Rituximab, n (%) |
2 (5.7) |
0 (0.0) |
0.151 |
|
JAK inhibitor, n (%) |
0 (0.0) |
2 (5.7) |
0.151 |
|
Corticosteroids, n (%) |
22 (62.9) |
6 (17.1) |
0.001 |
|
Nintedanib, n (%) |
1 (2.9) |
0 (0.0) |
0.314 |
|
Dose of corticosteroid, mg/d, mean (SD) |
7.9 (4.0) |
5.0 (0.0) |
0.135 |
|
Pulmonary function tests |
|
|
|
|
FVC <80%, n (%) |
28 (80.0) |
5 (14.3) |
<0.001 |
|
FVC predicted (%), mean (SD) |
63.0 (17.1) |
83.4 (4.4) |
<0.001 |
|
FEV1 <80%, n (%) |
23 (67.6) |
5 (14.3) |
0.001 |
|
FEV1 predicted (%), mean (SD) |
68.7 (15.9) |
84.0 (11.5) |
<0.001 |
|
DLCO <80%, n (%) |
29 (85.3) |
3 (8.6) |
<0.001 |
|
DLCO-SB predicted (%), mean (SD) |
61.0 (15.2) |
85.9 (7.9) |
<0.001 |
|
HRCT pattern |
|
|
|
|
UIP, n (%) |
29 (82.9) |
0 (0.0) |
<0.001 |
|
NSIP, n (%) |
6 (17.1) |
0 (0.0) |
<0.001 |
Abbreviations. RA: rheumatoid arthritis; ILD: interstitial lung disease; SD: standard deviation; RF: rheumatoid factor; ACPA: anti–citrullinated peptide antibody; csDMARD: conventional synthetic disease-modifying antirheumatic drug; bDMARD: biologic disease-modifying antirheumatic drug; FVC: forced vital capacity; FEV1: forced expiratory volume in the first second; DLCO: diffusing capacity of the lung for carbon monoxide; HRCT: high-resolution computed tomography; UIP: usual interstitial pneumonia; NSIP: nonspecific interstitial pneumonia.
Table S3. Baseline characteristics of 35 patients with RA-ILD and progression of lung disease.
|
VARIABLE |
RA-ILD with progression N = 13 |
RA-ILD without progression N = 22 |
p Value |
|
Epidemiologic characteristics |
|
|
|
|
Age, years, mean (SD) |
69.6 (9.2) |
69.8 (9.6) |
0.949 |
|
Male sex, n (%) |
8 (61.5) |
7 (31.8) |
0.086 |
|
Clinical-analytical characteristics |
|
|
|
|
Smoking history |
|
|
0.845 |
|
Nonsmokers, n (%) |
6 (46.2) |
10 (45.5) |
|
|
Ex-smokers, n (%) |
4 (30.8) |
7 (31.8) |
|
|
Active smokers, n (%) |
3 (23.1) |
5 (22.7) |
|
|
Duration of RA, months, median (IQR) |
180.7 (108.0-254.7) |
137.6 (79.5-244.6) |
0.511 |
|
Duration of ILD, months, mean (SD) |
83.3 (40.0) |
59.0 (49.0) |
0.100 |
|
RF+ (>10 IU), n (%) |
13 (100.0) |
19 (86.4) |
0.263 |
|
High RF (>60 IU) |
10 (76.9) |
14 (63.6) |
0.413 |
|
ACPA+ (>20 IU), n (%) |
13 (100.0) |
19 (86.4) |
0.164 |
|
ACPA titer, median (IQR) |
1010.0 (297.0-1682.0) |
415.5 (82.7-1015.5) |
0.098 |
|
High ACPA titer (>340 IU), n (%) |
11 (84.6) |
11 (50.0) |
0.041 |
|
Radiographic erosions, n (%) |
8 (61.6) |
13 (59.0) |
0.660 |
|
Clinical manifestations |
|
|
|
|
DAS28-ESR, mean (SD) |
3.2 (1.0) |
3.1 (0.9) |
0.602 |
|
Remission-low disease activity, n (%) |
6 (46.2) |
13 (59.1) |
0.347 |
|
Moderate-high disease activity, n (%) |
7 (53.8) |
9 (40.9) |
0.347 |
|
HAQ, mean (SD) |
1.3 (0.7) |
1.2 (0.6) |
0.511 |
|
Current treatment |
|
|
|
|
csDMARD, n (%) |
11 (84.6) |
17 (77.3) |
0.600 |
|
Methotrexate, n (%) |
8 (61.5) |
11 (50.0) |
0.508 |
|
Leflunomide, n (%) |
0 (0.0) |
3 (13.6) |
0.164 |
|
Sulfasalazine, n (%) |
2 (15.4) |
0 (0.0) |
0.058 |
|
Hydroxychloroquine, n (%) |
2 (15.4) |
4 (18.2) |
0.832 |
|
Mycophenolate, n (%) |
2 (15.4) |
2 (15.4) |
0.572 |
|
bDMARD, n (%) |
8 (61.5) |
14 (63.6) |
0.901 |
|
Anti-TNF, n (%) |
2 (15.4) |
2 (15.4) |
0.572 |
|
Tocilizumab, n (%) |
2 (15.4) |
1 (4.5) |
0.268 |
|
Abatacept, n (%) |
4 (30.8) |
9 (40.9) |
0.549 |
|
Rituximab, n (%) |
0 (0.0) |
2 (9.1) |
0.263 |
|
Corticosteroids, n (%) |
7 (53.8) |
15 (68.2) |
0.396 |
|
Dose of corticosteroids, mg/d mean (SD) |
6.6 (2.8) |
7.9 (4.0) |
0.649 |
|
Pulmonary function testing |
|
|
|
|
FVC <80%, n (%) |
13 (100.0) |
15 (68.2) |
0.031 |
|
FVC mean (SD) |
56.1 (17.3) |
67.3 (15.8) |
0.056 |
|
FEV1 <80%, n (%) |
10 (76.9) |
13 (60.0) |
0.201 |
|
FEV1 mean (SD) |
64.3 (17.1) |
71.5 (14.9) |
0.207 |
|
DLCO <80%, n (%) |
13 (100.0) |
16 (72.7) |
0.039 |
|
DLCO, mean (SD) |
62.9 (13.9) |
56.6 (17.7) |
0.049 |
|
HCRT radiological pattern |
|
|
|
|
UIP, n (%) |
12 (92.3) |
17 (77.3) |
0.254 |
|
NSIP, n (%) |
1 (7.7) |
5 (22.7) |
0.254 |
Abbreviations. RA: rheumatoid arthritis; ILD: interstitial lung disease; SD: standard deviation; RF: rheumatoid factor; ACPA: anti–citrullinated peptide antibodies; csDMARD: conventional synthetic disease-modifying antirheumatic drug; bDMARD: biologic disease-modifying antirheumatic drug; FVC: forced vital capacity; FEV1: forced expiratory volume in the first second; DLCO: diffusing capacity of the lung for carbon monoxide; HRCT: high-resolution computed tomography; UIP: usual interstitial pneumonia; NSIP: nonspecific interstitial pneumonia.
- Line 286-287 The sensitivity and specificity of ACPA for RA are high, however, only these have the same predictive capacity in ILD?
Reply: We appreciate your clarification. We have found that patients with high ACPA titers had a higher frequency of lung disease progression (p=0.041), and we have added this information to supplementary table 3. However, in our model, the only variable that was significantly associated with the progression of RA-ILD was IL-18 pg/mL.
Table S3. Baseline characteristics of 35 patients with RA-ILD and progression of lung disease.
|
VARIABLE |
RA-ILD with progression N = 13 |
RA-ILD without progression N = 22 |
p Value |
|
Epidemiologic characteristics |
|
|
|
|
Age, years, mean (SD) |
69.6 (9.2) |
69.8 (9.6) |
0.949 |
|
Male sex, n (%) |
8 (61.5) |
7 (31.8) |
0.086 |
|
Clinical-analytical characteristics |
|
|
|
|
Smoking history |
|
|
0.845 |
|
Nonsmokers, n (%) |
6 (46.2) |
10 (45.5) |
|
|
Ex-smokers, n (%) |
4 (30.8) |
7 (31.8) |
|
|
Active smokers, n (%) |
3 (23.1) |
5 (22.7) |
|
|
Duration of RA, months, median (IQR) |
180.7 (108.0-254.7) |
137.6 (79.5-244.6) |
0.511 |
|
Duration of ILD, months, mean (SD) |
83.3 (40.0) |
59.0 (49.0) |
0.100 |
|
RF+ (>10 IU), n (%) |
13 (100.0) |
19 (86.4) |
0.263 |
|
High RF (>60 IU) |
10 (76.9) |
14 (63.6) |
0.413 |
|
ACPA+ (>20 IU), n (%) |
13 (100.0) |
19 (86.4) |
0.164 |
|
ACPA titer, median (IQR) |
1010.0 (297.0-1682.0) |
415.5 (82.7-1015.5) |
0.098 |
|
High ACPA titer (>340 IU), n (%) |
11 (84.6) |
11 (50.0) |
0.041 |
|
Radiographic erosions, n (%) |
8 (61.6) |
13 (59.0) |
0.660 |
|
Clinical manifestations |
|
|
|
|
DAS28-ESR, mean (SD) |
3.2 (1.0) |
3.1 (0.9) |
0.602 |
|
Remission-low disease activity, n (%) |
6 (46.2) |
13 (59.1) |
0.347 |
|
Moderate-high disease activity, n (%) |
7 (53.8) |
9 (40.9) |
0.347 |
|
HAQ, mean (SD) |
1.3 (0.7) |
1.2 (0.6) |
0.511 |
|
Current treatment |
|
|
|
|
csDMARD, n (%) |
11 (84.6) |
17 (77.3) |
0.600 |
|
Methotrexate, n (%) |
8 (61.5) |
11 (50.0) |
0.508 |
|
Leflunomide, n (%) |
0 (0.0) |
3 (13.6) |
0.164 |
|
Sulfasalazine, n (%) |
2 (15.4) |
0 (0.0) |
0.058 |
|
Hydroxychloroquine, n (%) |
2 (15.4) |
4 (18.2) |
0.832 |
|
Mycophenolate, n (%) |
2 (15.4) |
2 (15.4) |
0.572 |
|
bDMARD, n (%) |
8 (61.5) |
14 (63.6) |
0.901 |
|
Anti-TNF, n (%) |
2 (15.4) |
2 (15.4) |
0.572 |
|
Tocilizumab, n (%) |
2 (15.4) |
1 (4.5) |
0.268 |
|
Abatacept, n (%) |
4 (30.8) |
9 (40.9) |
0.549 |
|
Rituximab, n (%) |
0 (0.0) |
2 (9.1) |
0.263 |
|
Corticosteroids, n (%) |
7 (53.8) |
15 (68.2) |
0.396 |
|
Dose of corticosteroids, mg/d mean (SD) |
6.6 (2.8) |
7.9 (4.0) |
0.649 |
|
Pulmonary function testing |
|
|
|
|
FVC <80%, n (%) |
13 (100.0) |
15 (68.2) |
0.031 |
|
FVC mean (SD) |
56.1 (17.3) |
67.3 (15.8) |
0.056 |
|
FEV1 <80%, n (%) |
10 (76.9) |
13 (60.0) |
0.201 |
|
FEV1 mean (SD) |
64.3 (17.1) |
71.5 (14.9) |
0.207 |
|
DLCO <80%, n (%) |
13 (100.0) |
16 (72.7) |
0.039 |
|
DLCO, mean (SD) |
62.9 (13.9) |
56.6 (17.7) |
0.049 |
|
HCRT radiological pattern |
|
|
|
|
UIP, n (%) |
12 (92.3) |
17 (77.3) |
0.254 |
|
NSIP, n (%) |
1 (7.7) |
5 (22.7) |
0.254 |
Abbreviations. RA: rheumatoid arthritis; ILD: interstitial lung disease; SD: standard deviation; RF: rheumatoid factor; ACPA: anti–citrullinated peptide antibodies; csDMARD: conventional synthetic disease-modifying antirheumatic drug; bDMARD: biologic disease-modifying antirheumatic drug; FVC: forced vital capacity; FEV1: forced expiratory volume in the first second; DLCO: diffusing capacity of the lung for carbon monoxide; HRCT: high-resolution computed tomography; UIP: usual interstitial pneumonia; NSIP: nonspecific interstitial pneumon
Reviewer #2: Dear Authors
Thank you for your manuscript submission. Your brilliant work is well-designed and well-presented. However, I recommend you to do some Minor Revisions as below:
- It is recommended to add a schematic figure showing the Materials and Methods procedures together with the Results obtained from your study. This figure will show all the effective procedures and outcomes to the readers, at once.
Reply: Following the reviewer's recommendations, we have included a figure showing the materials and methods procedures together with the results obtained from our study. We appreciate your comment, as it has improved the document.
Figure 2: Methods and results
- As you know, interleukins and toll-like receptors are sisters and handle pivotal activities in human immune system. Therefore, I suggest you to mention TLRs in Introduction section in brief. In this regard, it is recommended to read and add the following paper to References section of the manuscript to have a fruitful Introduction section:
-The Interleukin-1 (IL-1) Superfamily Cytokines and Their Single Nucleotide Polymorphisms (SNPs). J Immunol Res. 2022 Mar 26;2022:2054431. doi: 10.1155/2022/2054431. PMID: 35378905; PMCID: PMC8976653.
Reply: We thank the reviewer for showing us this very interesting work. We have added it to the introduction and discussion of our work. We think this has improved our review.
-Page 2; lines 92-95: “Interleukins are involved in pro and anti-inflammatory responses by their interaction with a wide range of receptors, e.g., Toll-like receptors (TLRs). Interleukins and TLRs are involved in cancers, infectious and autoimmune diseases (26).”
-Page 12; lines 347-361: “In the present study, we also found that IL-18 values were higher in cases than in controls. Similarly, this was the only cytokine associated with progression of RA-ILD. IL-18 is a member of the IL-1 cytokine superfamily and is produced predominantly by macrophages. Data from animal studies show that IL-18 can lead to pulmonary inflammation (52); in humans, IL-18 levels are increased in patients with idiopathic pulmonary fibrosis (53), patients with ILD-associated inflammatory myopathy (54), and patients with RA-ILD (8). Similarly, IL-18 levels have also been associated with the pathogenesis of RA and has high biologic activity in synovial fluid and sera of patients with RA (26,55). In this sense, Matsuo et al. recently showed that patients with RA-ILD had higher IL-18 values than RA patients without ILD; IL-8 was also associated with ILD, independently of inflammatory factors (8). This IL-18–mediated effect could be due to various mechanisms, i.e. a key role in polarization of Th1 cells, production of inflammatory cytokines by different cell strains in innate and acquired immunity, and differentiation of Th17 cells (56). While further studies are necessary to determine the exact role of this cytokine in RA-ILD, we might think that its association with greater inflammation and severity in RA is associated with the poorer outcome of ILD in affected patients. “
Reference:
- Behzadi P, Sameer AS, Nissar S, Banday MZ, Gajdács M, García-Perdomo HA, et al. The Interleukin-1 (IL-1) Superfamily Cytokines and Their Single Nucleotide Polymorphisms (SNPs). J Immunol Res. 2022;2022:2054431.
- Furthermore, it is recommended to read and add the following papers to References section of the manuscript to have effective Discussion section:
Reply: We appreciate the reviewer's comment and have included the bibliography.
-Novel Biomarkers, Diagnostic and Therapeutic Approach in Rheumatoid Arthritis Interstitial Lung Disease-A Narrative Review. Biomedicines. 2022 Jun 9;10(6):1367. doi: 10.3390/biomedicines10061367. PMID: 35740390; PMCID: PMC9219939.
-Serum Uric Acid as a Diagnostic Biomarker for Rheumatoid Arthritis-Associated Interstitial Lung Disease. Inflammation. 2022 Aug;45(4):1800-1814. doi: 10.1007/s10753-022-01661-w. Epub 2022 Mar 22. PMID: 35314903; PMCID: -PMC9197871.
-Predicting rheumatoid arthritis-associated interstitial lung disease: filling the void. The Lancet Rheumatology. 2023 Feb 1;5(2):e61-3
-Identification of biomarkers by machine learning classifiers to assist diagnose rheumatoid arthritis-associated interstitial lung disease. Arthritis Res Ther. 2022 May 19;24(1):115. doi: 10.1186/s13075-022-02800-2. PMID: 35590341; PMCID: PMC9118651.
-Recent Advances in Basic and Clinical Aspects of Rheumatoid Arthritis-associated Interstitial Lung Diseases. J Rheum Dis 2022;29:61-70. https://doi.org/10.4078/jrd.2022.29.2.61
-Development of a Risk Nomogram Model for Identifying Interstitial Lung Disease in Patients With Rheumatoid Arthritis. Front Immunol. 2022 Jun 16;13:823669. doi: 10.3389/fimmu.2022.823669. PMID: 35784288; PMCID: PMC9245420.
-Page 2; lines 64-66: “No useful serum biomarkers are currently available for diagnosis of RA-ILD, although various candidates have been evaluated. The protein Krebs von den Lungen 6 (KL-6) in serum has been investigated for diagnosis of ILD associated with systemic inflammatory diseases (5).”
-Page 2; lines 72-73: “Other studies have described that the elevated uric acid levels, D-dimer, and tumor markers may be a diagnostic marker in RA-ILD (13,14).”
-Page 2; lines 70-73: “Similarly, they have not proven to have greater predictive value than anti–citrullinated peptide antibody (ACPA) (11,12).”
-Page 2; lines 87-89: “Similarly, TGF beta1 (24) and IL-18 (8) have been associated with diagnosis and severity of RA-ILD (24), and the level of circulating MMP-3 increased the diagnostic accuracy for ILD in RA patients (25).”
References
5.Florescu A, Gherghina FL, Mușetescu AE, Pădureanu V, Roșu A, Florescu MM, et al. Novel Biomarkers, Diagnostic and Therapeutic Approach in Rheumatoid Arthritis Interstitial Lung Disease-A Narrative Review. Biomedicines. 2022 Jun;10(6).
- Kronzer VL, Hayashi K, Yoshida K, Davis JM 3rd, McDermott GC, Huang W, et al. Autoantibodies against citrullinated and native proteins and prediction of rheumatoid arthritis-associated interstitial lung disease: A nested case-control study. Lancet Rheumatol. 2023 Feb;5(2):e77–87.
- Wang Z, Wang W, Xiang T, Gong B, Xie J. Serum Uric Acid as a Diagnostic Biomarker for
Rheumatoid Arthritis-Associated Interstitial Lung Disease. Inflammation. 2022
Aug;45(4):1800–14.
- Qin Y, Wang Y, Meng F, Feng M, Zhao X, Gao C, et al. Identification of biomarkers by machine learning classifiers to assist diagnose rheumatoid arthritis-associated interstitial lung disease. Arthritis Res Ther. 2022 May;24(1):115.
- Xue J, Hu W, Wu S, Wang J, Chi S, Liu X. Development of a Risk Nomogram Model for Identifying Interstitial Lung Disease in Patients With Rheumatoid Arthritis. Front Immunol. 2022;13:823669.
The manuscript has been edited and checked by a native English-speaking funded by the Spanish Foundation of Rheumatology.
Page 17; lines 431-432, Acknowledgments: FERBT2021- The authors thank the Spanish Foundation of Rheumatology for providing medical writing/editorial assistance during the preparation of the manuscript.
Thank you in advance for your time and consideration.
Sincerely yours,
*Correspondence: Natalia Mena Vázquez MD, PhD.
Affiliation: UGC de Reumatología, Instituto de Investigación Biomédica de Málaga (IBIMA)-Plataforma Bionand, Hospital Regional Universitario de Málaga, Málaga, Spain. Plaza del Hospital Civil s/n., 29009 Malaga, Spain. E-mail: nataliamenavazquez@gmail.com
Telephone number/ Fax number: +34 951 290360

Reviewer 2 Report
Dear Authors
Thank you for your manuscript submission. Your brilliant work is well-designed and well-presented. However, I recommend you to do some Minor Revisions as below:
1. It is recommended to add a schematic figure showing the Materials and Methods procedures together with the Results obtained from your study. This figure will show all the effective procedures and outcomes to the readers, at once.
2. As you know, interleukins and toll-like receptors are sisters and handle pivotal activities in human immune system. Therefore, I suggest you to mention TLRs in Introduction section in brief. In this regard, it is recommended to read and add the following paper to References section of the manuscript to have a fruitful Introduction section:
The Interleukin-1 (IL-1) Superfamily Cytokines and Their Single Nucleotide Polymorphisms (SNPs). J Immunol Res. 2022 Mar 26;2022:2054431. doi: 10.1155/2022/2054431. PMID: 35378905; PMCID: PMC8976653.
3. Furthermore, it is recommended to read and add the following papers to References section of the manuscript to have effective Discussion section:
Novel Biomarkers, Diagnostic and Therapeutic Approach in Rheumatoid Arthritis Interstitial Lung Disease-A Narrative Review. Biomedicines. 2022 Jun 9;10(6):1367. doi: 10.3390/biomedicines10061367. PMID: 35740390; PMCID: PMC9219939.
Serum Uric Acid as a Diagnostic Biomarker for Rheumatoid Arthritis-Associated Interstitial Lung Disease. Inflammation. 2022 Aug;45(4):1800-1814. doi: 10.1007/s10753-022-01661-w. Epub 2022 Mar 22. PMID: 35314903; PMCID: PMC9197871.
Predicting rheumatoid arthritis-associated interstitial lung disease: filling the void. The Lancet Rheumatology. 2023 Feb 1;5(2):e61-3
Identification of biomarkers by machine learning classifiers to assist diagnose rheumatoid arthritis-associated interstitial lung disease. Arthritis Res Ther. 2022 May 19;24(1):115. doi: 10.1186/s13075-022-02800-2. PMID: 35590341; PMCID: PMC9118651.
Recent Advances in Basic and Clinical Aspects of Rheumatoid Arthritis-associated Interstitial Lung Diseases. J Rheum Dis 2022;29:61-70. https://doi.org/10.4078/jrd.2022.29.2.61
Development of a Risk Nomogram Model for Identifying Interstitial Lung Disease in Patients With Rheumatoid Arthritis. Front Immunol. 2022 Jun 16;13:823669. doi: 10.3389/fimmu.2022.823669. PMID: 35784288; PMCID: PMC9245420.
Author Response

(The authors gave the same response as above.)
